# Neuroprotective Effects of the DPP4 Inhibitor Vildagliptin in In Vivo and In Vitro Models of Parkinson’s Disease

**DOI:** 10.3390/ijms23042388

**Published:** 2022-02-21

**Authors:** Ramesh Pariyar, Tonking Bastola, Dae Ho Lee, Jungwon Seo

**Affiliations:** 1Institute of Pharmaceutical Research and Development, College of Pharmacy, Wonkwang University, Iksan 54538, Korea; ume.ramesh@gmail.com (R.P.); tonkingbastola@gmail.com (T.B.); 2Department of Neuroscience, Cell Biology, and Anatomy, University of Texas Medical Branch, Galveston, TX 77555-0625, USA; 3Department of Internal Medicine, Gil Medical Center, Gachon University College of Medicine, Incheon 21565, Korea; drhormone@naver.com

**Keywords:** vildagliptin, MPTP, Parkinson’s disease, apoptosis, dopaminergic neuron, motor dysfunction, autophagy

## Abstract

Parkinson’s disease (PD) is characterized by loss of dopaminergic neurons in the substantia nigra pars compacta (SNpc) of the midbrain. Restoration of nigrostriatal dopamine neurons has been proposed as a potential therapeutic strategy for PD. Because currently used PD therapeutics only help relieve motor symptoms and do not treat the cause of the disease, highly effective drugs are needed. Vildagliptin, a dipeptidyl peptidase 4 (DPP4) inhibitor, is an anti-diabetic drug with various pharmacological properties including neuroprotective effects. However, the detailed effects of vildagliptin against PD are not fully understood. We investigated the effects of vildagliptin on PD and its underlying molecular mechanisms using a 1-methyl-4-phenyl-1,2,3,6-tetrahydropyridine (MPTP)-induced mouse model and a 1-methyl-4-phenylpyridium (MPP^+^)-induced cytotoxicity model. Vildagliptin (50 mg/kg) administration significantly attenuated MPTP-induced motor deficits as evidenced by rotarod, pole, and nest building tests. Immunohistochemistry and Western blot analysis revealed that vildagliptin increased tyrosine hydroxylase-positive cells in the SNpc and striatum, which was reduced by MPTP treatment. Furthermore, vildagliptin activated MPTP-decreased PI3k/Akt and mitigated MPTP-increased ERK and JNK signaling pathways in the striatum. Consistent with signaling transduction in the mouse striatum, vildagliptin reversed MPP^+^-induced dephosphorylation of PI3K/Akt and phosphorylation of ERK and JNK in SH-SY5Y cells. Moreover, vildagliptin attenuated MPP^+^-induced conversion of LC3B-II in SH-SY5Y cells, suggesting its role in autophagy inhibition. Taken together, these findings indicate that vildagliptin has protective effects against MPTP-induced motor dysfunction by inhibiting dopaminergic neuronal apoptosis, which is associated with regulation of PI3k/Akt, ERK, and JNK signaling transduction. Our findings suggest vildagliptin as a promising repurposing drug to treat PD.

## 1. Introduction

Parkinson’s disease (PD) is the second most common neurodegenerative disorder after Alzheimer’s disease [1]. Clinically, PD symptoms are divided into two categories, motor and non-motor symptoms. The motor symptoms are resting tremor, bradykinesia, rigidity, and posture instability [2,3]. The main pathological hallmark of PD is progressive death of dopaminergic neurons in the substantia nigra pars compacta (SNpc) and striatum in the brain. The pathology responsible for the clinical conditions is accompanied by degeneration of dopaminergic neurons in the SNpc [4] and thereby reducing dopamine level in the striatum [5,6]. Cardinal symptoms of PD are observed only after the dopamine level in the striatum is decreased by 60–80% [7]. Therefore, restoring dopamine/acetylcholine balance is the primary goal of pharmacotherapy treatment to mitigate symptoms associated with PD [8]. Recently, real-time monitoring of dopamine release based on nanocomposites has been studied as a potential diagnostic tool for PD [9,10].

For development of PD therapeutics, a 1-methyl-4-phenyl-1,2,3,6-tetrahydropyridine (MPTP)-treated mouse model has been widely used [11]. In particular, MPTP induces selective neurotoxicity in the nigrostriatal dopaminergic neurons of the mouse brain, which provides a useful tool for researching PD [12]. 1-Methyl-4-phenylpyridium (MPP^+^), metabolized from MPTP by the enzyme monoamine oxidase-B, decreases striatum dopamine and increases nigrostriatal neuronal loss and behavioral impairments [13,14,15]. Although it has the limitation of not fully mimicking PD symptoms, the MPTP-induced mouse model has the advantages of simplicity, practicality, and clinical correlation compared to other toxin models.

Dopaminergic medications, such as levodopa, ameliorate the motor symptoms in the early stages of PD. However, after a few years, patients can develop dopa-resistant symptoms and several side effects of such medications [16]. Therefore, it is necessary to develop novel PD drugs with long-term efficacy and safety. For PD drug development, drug repurposing is an effective strategy, in which existing safe medicines are inexpensively repositioned based on valid PD target molecules.

Recently, antidiabetic dipeptidyl peptidase 4 (DPP4) inhibitors were reported to have beneficial effects on diabetic patients with PD [17]. In the report, diabetic patients with PD being treated with DPP4 inhibitors had higher dopamine transporter availability in the putamina and a lower rate of levodopa-induced dyskinesia than that in diabetic patients with PD not taking DPP4 inhibitors. Furthermore, diabetic PD patients treated with DPP4 inhibitors had higher dopamine transporter availability in the posterior putamen than that in non-diabetic PD patients. Therefore, it is likely that DPP4 inhibitors have potential for treating PD, particularly in diabetic patients with PD. DPP4 is a serine exopeptidase enzyme responsible for degradation of incretins such as glucagon-like peptide-1 (GLP-1), which reduces the blood glucose level by stimulating insulin secretion [18]. DPP4 inhibition also modulates the activity of other factors with potential neuroprotective properties [19]. Vildagliptin, a DPP4 inhibitor, is a novel anti-hyperglycemic drug with few adverse effects [20]. Recently, vildagliptin was reported to show neuroprotective effects against rotenone-induced toxicity through its anti-apoptosis and anti-oxidant properties [21]. In addition, a neuroprotective effect of vildagliptin was demonstrated against cerebral ischemia [22]. However, there are no findings on the detailed effects of vildagliptin to protect against MPTP/MPP^+^-induced PD models. In order to explore its potential as a repurposing drug for PD, we examined whether vildagliptin could ameliorate the motor symptoms in an MPTP-induced mouse model. In addition, we investigated the detailed molecular mechanism of vildagliptin effects using the MPTP-induced mouse model and the MPP^+^-induced neurotoxicity model.

## 2. Results

### 2.1. Vildagliptin Ameliorates Behavior Performance in MPTP-Induced Motor Impairments

The cardinal motor functions are tested as the evaluation index of PD [23]. To examine motor functions in our study, we used three behavior tests after vildagliptin and/or MPTP treatment as shown in Figure 1A. Test mice were orally administered 30 or 50 mg/kg vildagliptin for one week and then co-administered MPTP (30 mg/kg, *i.p.*) for another one week. Then, the rotarod test measuring latency to fall from a rotarod apparatus was performed to evaluate motor coordination and motor learning (Figure 1B). Remarkable motor dysfunction was observed in MPTP-injected mice, as determined by the decreased latency time compared to the control group. Pretreatment with 50 mg/kg vildagliptin before MPTP injection significantly increased latency to fall, indicating motor function rescue. The latency time of the 30 mg/kg vildagliptin and MPTP group was improved compared to that of MPTP group but with no significant difference. In addition, vildagliptin alone showed no significant changes in rotarod performance compared to the control. These data suggest that the motor coordination deficits caused by MPTP are alleviated by prophylactic treatment with vildagliptin.

Bradykinesia occurs in the majority of PD patients and occurred in the MPTP-induced PD mouse model [24]. Therefore, we implemented the pole test to measure bradykinesia caused by depletion of striatal dopamine. The total time required to climb from the top of the pole to the ground was evaluated. As shown in Figure 1C, the total locomotor activity time was significantly prolonged after MPTP treatment. Pretreatment with 50 mg/kg vildagliptin significantly shortened the time needed to reach the platform, suggesting that vildagliptin prevented MPTP-induced bradykinesia. The vildagliptin (30 mg/kg) and MPTP group showed a positive effect compared to that in the MPTP group, but the difference was not significant. Based on these results, we selected the 50 mg/kg vildagliptin and MPTP group for our further experiments.

Nest building behavior requires orofacial and forelimb movement, which are dopamine dependent [25]. To assess nest building performance, each mouse was housed in a cage containing a single block of nesting cotton for 18 h. Each cage was then scored blindly on a scale ranging from 1 (non-shredded cotton) to 5 (maximum shredded cotton) depending on the condition of the nesting material [26]. The nesting material in the MPTP group was less shredded than that in the control group, shown as a decreased nesting score (Figure 1D). Mice in the vildagliptin (50 mg/kg) and MPTP group showed an increased nesting score compared to that in the MPTP group.

All these behavior tests suggest that the dopamine-dependent motor deficits caused by MPTP are alleviated by pretreatment with vildagliptin.

### 2.2. Vildagliptin Alleviates MPTP-Induced Loss of Tyrosine Hydroxylase-Positive Neurons in the SNpc and Striatum

MPTP-induced motor dysfunction results from loss of dopaminergic neurons in the SNpc and striatum [27]. To investigate the effect of vildagliptin against MPTP-induced damage in dopaminergic neurons, tyrosine hydroxylase (TH)-immunoreactive cells in the SNpc and striatum of mice were quantified. Consistent with previous reports [28,29], MPTP induced a severe reduction in TH-positive cells in the SNpc of mice (Figure 2A). Vildagliptin pretreatment markedly increased TH-immunoreactive cells in the SNpc region compared to those in the MPTP group, suggesting that vildagliptin has protective effects against MPTP-induced degeneration of dopaminergic neurons. In the striatum, MPTP treatment reduced the number of TH-immunoreactive neurofilaments, while vildagliptin pretreatment greatly hindered this loss of TH expression (Figure 2B). Moreover, no difference in TH-immunoreactive neurons was observed between the vildagliptin alone-treated group and the control group. TH protein expression in the SNpc was also analyzed by immunoblotting assay and was quantified (Figure 2C). MPTP treatment reduced the level of TH expression compared to that in the control, whereas vildagliptin treatment markedly prevented the decrease in TH level by MPTP.

Taken together, these findings show that vildagliptin pretreatment alleviated the reduction in TH-positive cells by MPTP in the SNpc and striatum, suggesting that vildagliptin protects dopaminergic neurons from MPTP neurotoxicity in the mouse brain.

### 2.3. Vildagliptin Attenuates MPTP-Induced Apoptosis in the Striatum

Apoptosis is an important mechanism in dopaminergic neuron loss in PD and is modeled in MPTP-treated mice [30]. We performed Western blot analysis to examine the expression of apoptosis markers of Bax, Bcl2, and caspase-3 in the striatum of mouse brains treated with MPTP and/or vildagliptin (Figure 3). We found that vildagliptin significantly reduced the MPTP-induced cleavage of caspase-3. Consistently, MPTP decreased Bcl2 expression but increased Bax expression, leading to a significantly increased Bax/Bcl2 ratio. On the contrary, vildagliptin restored the Bax/Bcl2 ratio increased by MPTP treatment. These results indicate that vildagliptin prevents MPTP-induced neuronal apoptosis in the striatum of the mouse brain.

### 2.4. MPTP Modifies PI3K/Akt and MAPK Signaling Pathways, Which Is Prevented by Vildagliptin Pretreatment in the Striatum

Next, we identified the molecular mechanism underlying the vildagliptin-induced protective effect against MPTP-induced apoptosis in mouse striatum. We assessed the neuronal survival signaling pathway of intracellular phosphoinositide 3-kinase inhibitor (PI3K)/Akt and brain-derived neurotrophic factor (BDNF) and the apoptotic signaling pathway of mitogen-activated protein kinases (MAPK). As shown in Figure 4A, Western blot analysis revealed that MPTP treatment inhibited PI3K/Akt signaling, as phosphorylation of Akt was reduced. However, the vildagliptin pretreatment group exhibited a significantly higher level of phosphorylated Akt. Next, we evaluated BDNF, which is a major neurotrophic factor involved in neuronal cell survival and dopamine release [31,32,33]. Vildagliptin treatment slightly increased the BDNF level, but there was no significant difference compared to that of the MPTP group. In addition, MPTP stimulated MAPK signaling, as evidenced by increased phosphorylation of ERK and JNK compared to that in the control group (Figure 4B). Vildagliptin pretreatment markedly hindered the elevated phosphorylation of ERK and JNK induced by MPTP. Vildagliptin alone did not affect the PI3K and MAPK signaling pathways compared with the control group. Our findings indicate that vildagliptin protects against MPTP-induced apoptosis of dopaminergic neurons in the SNpc by restoring the PI3K/Akt and MAPK signaling pathways.

### 2.5. Vildagliptin Attenuates MPP^+^-Induced Cell Death and Regulates the PI3K/Akt and MAPKs Signaling Pathways in SH-SY5Y Cells

To confirm the protective effects of vildagliptin against dopaminergic neuronal death, we administered 200 μM MPP^+^ with or without vildagliptin to SH-SY5Y cells for 48 h. First, we determined cell viability, as revealed by MTT assay (Figure 5A). MPP^+^ attenuated the viability of SH-SY5Y cells, while pretreatment with 5 or 10 μM vildagliptin showed a slight significant increase in viability against MPP^+^-induced toxicity. Furthermore, we evaluated the molecular signaling pathways assessed in the MPTP mouse model. MPP^+^ decreased the phosphorylation of Akt and increased the phosphorylation of ERK, JNK, and p38. Consistent with the in vivo effect, vildagliptin pretreatment prevented dephosphorylation of Akt and phosphorylation of ERK and JNK in SH-SY5Y cells (Figure 5B). These results support the neuroprotective molecular mechanism of vildagliptin.

### 2.6. Vildagliptin Inhibits MPP^+^-Induced Autophagy in SH-SY5Y Cells

Autophagy is a key mechanism in various physiopathological processes, including growth, development, cell death, and survival. Autophagy is one of the pathogeneses of PD, in which a defective process leads to an imbalance in homeostasis in neurons, resulting in an increase in the number of autophagosomes in the brain tissue [34]. Many recent studies have reported that autophagy is induced by MPP^+^ treatment in in vitro PD models [35,36]. Therefore, we examined whether vildagliptin regulated autophagy using the immunoblotting of the autophagy marker microtubule-associated protein 1A/1B-light chain 3B (LC3B) in MPP^+^-treated SH-SY5Y cells. As shown in Figure 6A, LC3B-II conversion was apparent in the MPP^+^-treated group, whereas vildagliptin pretreatment significantly hindered conversion of LC3B-II. These results imply that vildagliptin inhibited MPP^+^-induced dopaminergic neuronal autophagy. Next, we examined whether vildagliptin regulated autophagy in the SNpc and striatum of the MPTP model mouse (Figure 6B,C). However, we failed to find any alterations of LC3B expression and LC3B-II conversion with MPTP and/or vildagliptin.

## 3. Discussion

PD is a progressive neurodegenerative disease that affects motor function. The current therapeutics for PD can help manage the symptoms but do not slow or reverse the disease progression [37]. PD drugs also lead to drug-resistance symptoms in long-term therapy or to various side effects [16]. In order to attenuate side effects and provide further patient-specific care, novel safe drugs with various therapeutic mechanisms are needed. Drug repurposing is an efficient strategy for lower development costs and shorter development timelines [38]. Several lines of evidence have shown that vildagliptin possesses various biological functions including anti-diabetes, anti-oxidant, anti-inflammation, anti-apoptosis, and anti-cancer activities and cardiovascular protection [20,21,22,39,40,41,42]. In addition, recent evidence has indicated the beneficial action of vildagliptin in neurodegenerative diseases [21,22]. However, the effects of vildagliptin on the nigrostriatal dopaminergic system in a PD mouse model had not been studied. Therefore, the goal of this study was to evaluate the potential of vildagliptin as a repurposed drug for PD using an MPTP-induced PD mouse model and to examine the underlying molecular mechanisms.

Motor function is usually applied as the evaluation index of PD [43]. The rotarod test measuring the ability to balance and to walk on a rotating rod is well established to investigate motor function and coordination [44]. As shown in Figure 1B, MPTP significantly reduced the retention time on a rotarod compared to the control, while pretreatment with 50 mg/kg vildagliptin showed a significantly longer retention time compared to the MPTP group. To confirm motor deficits due to nigrostriatal dysfunction, the pole test was conducted. The pole test, which monitors forepaw dexterity of mice, was reported to be highly correlated with striatum dopamine content [45]. While the mice in the MPTP group took significantly more time to descend than those in the control group, vildagliptin pretreatment inhibited MPTP-induced bradykinesia (Figure 1C). Similarly, nest building behavior requires orofacial and forelimb movement in a dopamine-dependent manner, making it assess Parkinson-like symptoms [25]. The nesting score of the MPTP group was lower than that of the control group, whereas the mice in the MPTP and vildagliptin group showed significantly higher nest quality compared to the MPTP group (Figure 1D). All together, these behavior tests suggest that vildagliptin pretreatment hinders MPTP-induced motor dysfunction, reflecting its ability to maintain the nigrostriatal dopaminergic system.

The motor dysfunction in PD is mediated by degeneration of dopaminergic neurons in the SNpc and consequent dopamine depletion in the striatum [7]. PD patients show decreased dopamine level and reduced TH protein, especially in the nigrostriatal system [46]. MPTP selectively damages the dopamine-producing neurons in the SNpc, which decreases dopamine in the nigral part of the brain [47]. Therefore, we evaluated whether vildagliptin could protect dopaminergic neurons from MPTP toxicity in mouse brain. Immunohistochemical and Western blot assay revealed that TH expression was significantly reduced by MPTP treatment, indicating considerable nigrostriatal dopaminergic degeneration (Figure 2). TH is a rate-limiting enzyme of dopamine biosynthesis and 3,4-dihydroxyphenylalanine conversion. Vildagliptin pretreatment led to higher protein expression of TH in SNpc and striatum regions compared to the MPTP group, implicating protective effects of vildagliptin on dopaminergic neurons.

Herein, to elucidate the possible mechanism of the neuroprotective effects of vildagliptin, the expression of apoptosis-related proteins in striatum was determined. The apoptotic death of dopaminergic neurons plays an initial major role in progression of PD [30]. Several studies have shown caspase activation and apoptosis in PD patients and in animal models of PD [48,49]. In a previous report, vildagliptin attenuated mitochondria-mediated apoptosis and suppressed the oxidative stress in a rotenone-induced PD model [21]. Mitochondrial dysfunction is one of the main causes of neurodegenerative diseases and leads to release of cytochrome C into the cytoplasm, followed by activation of a caspase cascade [50]. MPTP and MPP^+^ activate caspase-9 and caspase-3 in both in vivo and in vitro models [50,51]. In line with these studies, we found significantly increased caspase-3 expression in the MPTP-treated group compared to that in the control group, whereas vildagliptin pretreatment inhibited the MPTP-induced activation of caspase-3 (Figure 3). In addition, the balance of pro-apoptotic and anti-apoptotic proteins in the Bcl2 family plays a central role in the regulation of caspase activation [52]. Although both Bax and Bcl2 belong to the Bcl2 family, they exert opposite functions, with pro-apoptosis and anti-apoptosis effects, respectively [53]. It is well established that MPTP/MPP^+^ alters the Bax/Bcl2 ratio in dopaminergic neurons [29,54,55]. In concordance with these studies, MPTP significantly increased the Bax/Bcl2 ratio in our experiment, while vildagliptin pretreatment markedly inhibited the increment in the ratio of Bax/Bcl2 (Figure 3), suggesting anti-apoptotic effects of vildagliptin on mitochondria-mediated neurotoxicity by MPTP.

One of the most important findings in this study is to elucidate the molecular mechanism underlying the therapeutic effects of vildagliptin against PD. To explore the upstream signaling pathway involved in the neuroprotective effect of vildagliptin, PI3K/Akt and MAPKs signaling pathways were investigated. Western blot analysis clearly showed that MPTP induced the dephosphorylation of Akt and the phosphorylation of JNK and ERK in the striatum. Furthermore, vildagliptin pretreatment significantly maintained phosphorylation of Akt, JNK and ERK compared with those in the MPTP-treated group (Figure 4A). Consistent with these in vivo results, vildagliptin hindered dephosphorylation of Akt and phosphorylation of ERK and JNK by MPP^+^ treatment in SH-SY5Y cells (Figure 5).

Studies have shown that activation of the PI3K/Akt signaling pathway is essential for protecting neuronal cells from oxidative stress [29,56]. Akt protein kinase is the PI3K signaling downstream effector, and phosphorylation of Akt is required for PI3K/Akt signaling transduction [57,58]. Akt consequently phosphorylates the transcription factors and regulates the pro-apoptosis and anti-apoptosis balance, thereby promoting neuronal survival [59]. Previous reports have demonstrated that delivery of a dominant-negative form of Akt decreased dopaminergic neurons and expression of a constitutively active-form reduced cell death of developmental neuron in the substantia nigra [60]. The MAPK signaling pathway, which regulates various physiological processes, including cell survival or cell death, is implicated in PD pathology [61]. In various PD models induced by neurotoxins such as MPTP or 6-hydroxydopamine, JNK signaling was shown to be significantly activated [62,63]. Furthermore, JNK inhibition by SP-600125 or SR-3306 inhibitors mitigates dopaminergic neuronal apoptosis in MPTP and/or 6-OHDA PD models in vivo and MPP^+^-induced neurotoxic PD models in vitro [64,65]. In addition, ERK1/2 signaling is associated with neuronal cell death in neurodegenerative diseases [66]. PD98059, an ERK1/2 pathway inhibitor, alleviated MPP^+^-induced apoptosis in differentiated PC12 cells [67] and H_2_O_2_-induced cell death in the oligodendroglial CG4 cells [68]. Another ERK1/2 inhibitor, U0126, also reversed dopamine-induced cell death of primary striatal neurons [69]. Both in vivo and in vitro results suggest that vildagliptin can prevent the MPTP/MPP^+^-induced dopaminergic neuronal cell death through regulation of the PI3K/Akt, JNKs, and ERK1/2 signaling pathways. In the p38 signaling pathway, MPP^+^ treatment significantly increased phosphorylation, but vildagliptin did not alter p38 phosphorylation, indicating selective regulation of vildagliptin on the MAPK signaling pathway. As shown in Figure 5A, vildagliptin did not fully maintain the cell viability in MPP^+^-treated SH-SY5Y cells, although the increase in viability was significant. This insufficient recovery might be associated with MPP^+^-increased p38 phosphorylation, which was not altered by vildagliptin. Further study will be needed to clarify this issue.

The impaired autophagy increases the number of autophagosomes in the brain tissue of PD patients [34] and PD-model animals induced by MPTP, 6-hydroxydopamine, or rotenone [70]. The autophagy-related protein LC3B, which is responsible for autophagosome formation, was immunopositive in the Lewy bodies of PD patients [71,72]. Especially, the lipidated form of LC3B (LC3B-II) was highly increased in the substantia nigra of PD brains. In cellular models, it has been reported that stimulated autophagy induces cell death in MPP^+^-treated human neuroblastoma SH-SY5Y cells [35,36,55,73]. Although it is not fully understood, the detailed mechanism of impaired autophagy in PD pathogenesis seems to be closely linked with dopaminergic neuron degeneration and α–synuclein accumulation [74]. As shown in Figure 6A, MPP^+^ induced LC3B-II conversion, accompanied by cell death. Vildagliptin significantly inhibited MPP^+^-induced LC3B-II conversion in SH-SY5Y cells, suggesting that vildagliptin regulates autophagy, which is impaired in the brains of PD patients. Therefore, autophagy changes were examined through detection of LC3B expression in the MPTP-treated mouse SNpc and striatum (Figure 6B,C). However, we failed to find alterations in LC3B expression and LC3B-II conversion after MPTP and/or vildagliptin treatment. To understand the role of autophagy associated with PD and the effects of vildagliptin on autophagy, it is necessary to collect brain tissues at different time points or to utilize different PD animal models. Considering the emerging importance of autophagy in PD pathogenesis, further investigation will be needed.

In conclusion, vildagliptin restored MPTP-induced dopaminergic neurodegeneration in the SNpc and striatum of the mouse brain and ameliorated MPTP-induced motor dysfunction. Vildagliptin also reduced MPP^+^-induced neurocytotoxicity in SH-SY5Y cells. Furthermore, vildagliptin inhibited MPTP/MPP^+^-induced dephosphorylation of Akt and phosphorylation of JNK and ERK in in vivo and in vitro PD models. Moreover, vildagliptin mitigated MPP^+^-induced autophagy in SH-SY5Y cells. Our data demonstrate that vildagliptin might have therapeutic effects in PD by inhibiting dopaminergic neuronal apoptosis in the nigrostriatal system, suggesting the potential of vildagliptin as a repurposed drug against PD.

## 4. Materials and Methods

### 4.1. Materials

MPTP, MPP^+^, vildagliptin, and DCFH-DA were purchased from Sigma Chemicals (St. Louis, MO, USA). Dulbecco’s modified Eagle medium (DMEM), fetal bovine serum (FBS), 0.5% Trypsin-EDTA, penicillin, and streptomycin were purchased from Gibco (Carlsbad, CA, USA). Dimethyl sulfoxide (DMSO) was purchased from Amresco (Solon, OH, USA). Anti-tyrosine hydroxylase antibody was purchased from Merck. Anti-phospho-Akt (Ser473), anti-Akt, anti-phospho-ERK1/2 (Thr202/Tyr204), anti-ERK, anti-phospho-JNK, anti-JNK, anti-phospho-p38, anti-p38, anti-Bax, anti-Bcl2, and anti-GAPDH antibodies were obtained from Cell Signaling Technology. The secondary antibodies were purchased from Santa Cruz Biotechnology (Santa Cruz, CA, USA). Human neuroblastoma SH-SY5Y cells were purchased from the Korean Cell Line Bank.

### 4.2. SH-SY5Y Cell Culture

SH-SY5Y cells were subcultured in DMEM and supplied with 10% FBS, 100 U/mL penicillin, and 100 U/mL streptomycin in a humidified atmosphere of 95% air and 5% CO_2_ at 37 °C. The cells were seeded in 96-well plates and 6-well plates at densities of 1 × 10^5^ and 2 × 10^6^ cells, respectively. Experiments were performed with cells at 70% confluence. Vildagliptin was dissolved in water.

### 4.3. Cell Viability Assay

Cell viability was measured using the 3-(4,5-dimethylthiazol-2-yl)-2,5-diphenyl tetrazolium bromide (MTT) assay, as previously reported [75]. In brief, SH-SY5Y cells (1 × 10^5^ cells/well) seeded in 96-well plates were pretreated with various concentrations of vildagliptin for 2 h and then incubated with 200 μM MPP^+^ for 48 h in 1% FBS media. Thereafter, MTT solution (1 mg/mL) was added into each well and incubated at room temperature for 2 h. The resulting MTT formazan was extracted with 100 μL of DMSO, and the absorbance was measured at 540 nm with a microtiter plate reader.

### 4.4. Animals

Adult male C57BL/6 mice weighing 25–30 g were purchased from Samtako (Osan, Korea). All mice were housed in a specific-pathogen-free area at 20 ± 3 °C with 55% humidity, a 12:12 h light:dark cycle, and ad libitum food and water. Mice were randomly allocated to five groups (*n* = 12 per group) of control, 50 mg/kg vildagliptin, MPTP, MPTP + 30 mg/kg vildagliptin, and MPTP + 50 mg/kg vildagliptin. Six mice in each group were used for the rotarod and nest building tests and immunohistochemistry. Another six mice were used for analysis of pole test and Western blotting analysis. All experiments on animals were approved by the Wonkwang University Institutional Animal Care and Use Committee (No. WKU16-23) and were performed in accordance with the guidelines of the National Institute of Toxicological Research of the Korea Food and Drug Administration.

### 4.5. MPTP Injection and Vildagliptin Administration

Mice were pretreated with vildagliptin (30 or 50 mg/kg/day, *p.o.*) for one week. Control and MPTP groups were pretreated with saline. After one week, the mice in the MPTP group and MPTP + vildagliptin group were injected with MPTP (30 mg/kg/day, *i.p.*) for another week. Control group mice were injected with saline on the same schedule. The behavior tests were performed after the last treatment of vildagliptin on the 14th day. Vildagliptin dose was prepared as an emulsion in distilled saline, and MPTP was first dissolved in DMSO and diluted in saline.

### 4.6. Behavioral Tests

#### 4.6.1. Rotarod Test

An accelerating rotarod (Ugo Basile, Italy) provides an index of fore and hind limb motor coordination and balance [76]. The rotarod is 3 cm in diameter and 30 cm long, with a non-slippery surface and 15 cm over the base of the apparatus. This rod is divided into 5 equal sections, allowing 5 mice to walk on the rod at the same time. The mice were trained 3 sessions per day for two consecutive days to adapt to the apparatus. For training and testing, the apparatus was set in accelerating mode from 4 to 40 rpm for 300 s. Baseline performance was measured before MPTP treatment. The test was performed after administration of MPTP and involved automatic records of when the mouse fell from the rotarod. The cutoff time was set at 300 s. The average retention time on the rod was calculated and plotted as a graph.

#### 4.6.2. Pole Test

The pole test is used to evaluate motor deficits and bradykinesia in mouse PD models [77]. MPTP influences motor coordination and balance of a mouse because of the dopaminergic neuron loss in the SNpc, prolonging the descent time from the pole [77]. Mice were placed head-down at the top of the pole (10 mm in diameter and 55 cm in height, with a rough surface), and the time to reach the bottom of the pole was measured. Mice were adapted to the pole for two days before MPTP treatment. The test was performed after administration of the last dose of MPTP. The test was repeated 3 times at 30 s intervals, and the locomotor activity time was evaluated as the average of all experimental values.

#### 4.6.3. Nest Building Test

Mice were individually housed overnight with fresh nesting material (cotton fiber) in a cage. On the next day (14th day of the experiment), before performing the pole test, nest building behavior was observed and scored by the blinded observers. Pictures of the nests were obtained from above before being scored by the blinded observers. The nests were observed and scored based on a rating from 1–5 points as follows: (1) nest material not noticeably touched, (2) nest material partially torn (50–90% remaining intact), (3) nest material mostly shredded (50–90% shredded), (4) identifiable but flat nest (more than 90% shredded), and (5) well-formed nest (more than 90% shredded) [26].

### 4.7. Tissue Collection

After the behavior test, six mice from each group were anesthetized with 2% urethane (sigma) and decapitated. The brain was removed from the skull, and the SNpc and striatum were collected. Another six mice from each group were anaesthetized with 2% urethane and perfused intracardially with PBS followed by 4% paraformaldehyde (PFA) in 0.1 M phosphate buffer (pH 7.4). Mouse brains were fixed in 4% PFA for 48 h and dehydrated in 30% sucrose solution in 0.1 M phosphate buffer (pH 7.4) for storage at 4 °C prior to sectioning. Brains were embedded with OCT (Sakura Finetek, Torrance, CA, USA) and cut into 30 μM thick sections on a Leica CM cryostat.

### 4.8. Immunohistochemistry Assay

The brain sections were immunostained with an RTU Elite ABC kit (Vector Laboratories, Burlingame, CA, USA). In brief, the sections were incubated in 0.3% hydrogen peroxide and washed in phosphate-buffered saline. After a 1 h incubation with blocking serum (2.5% horse serum), the sections were sequentially incubated with primary antibody against TH at 4 °C overnight, with the biotinylated secondary antibody for 2 h at room temperature, and then with an avidin–biotin–peroxidase complex mixture for 1 h. The antigen–antibody complex was visualized with 3,3’-diaminobenzidine chromogen (ImmPACT DAB Peroxidase Substrate Kit, Vector Laboratories). The sections were mounted, air-dried, dehydrated, cover-slipped, and observed under a TS100 light microscope (Nikon Instruments, Tokyo, Japan). Photomicrographs were obtained of the striatum and the SNpc regions at 40× and 100× magnification, respectively. The quantity of dopaminergic neurons was evaluated by counting TH-positive neurons in the SNpc in each group of mice, and the graph was plotted as fold-change relative to the control group. For optical density (OD) analysis of TH, immunohistochemically stained brain sections were imaged, and OD was analyzed using ImageJ. Images were converted into grayscale and inverted before analysis. Brain striatum area was selected and analyzed after subtracting the background staining value.

### 4.9. Western Blot Analysis

Tissue samples from SNpc or striatum were homogenized in ice-cold radioimmunoprecipitation assay (RIPA) buffer (150 mM NaCl, 1% Triton X-100, 1% sodium deoxycholate, 0.1% sodium dodecyl sulfate, 50 mM Tris-HCl, and 2 mM ethylenediaminetetraacetic acid) supplemented with a protease inhibitor cocktail and phosphatase inhibitors (Roche, Basel, Switzerland). The homogenate was centrifuged at 15,000 rpm/min for 10 min at 4 °C to remove debris. Protein concentrations were determined using a bicinchoninic acid assay kit (Thermo Fisher Scientific, Waltham, MA, USA). Equal quantities of cell lysate protein were loaded into SDS-PAGE (10–12%) gel. After electrophoresis, the results were transferred to polyvinylidene difluoride membranes (Millipore Corp., Burlington, MA, USA). After blocking in 5% skim milk, the membranes were incubated overnight at 4 °C with antibodies specific to phosphorylated Akt (Ser473), Akt, phosphorylated ERK, ERK, phosphorylated p38, p38, phosphorylated JNK, JNK, Bax, Bcl2, TH, and GAPDH; all primary antibodies were diluted 1:1000 in blocking solution. Next, the membranes were incubated with the corresponding secondary antibodies and developed using a chemiluminescent reagent (Thermo Scientific). The relative intensities of the specific protein bands were quantified using ImageJ (Version 1.51, National Institutes of Health) software.

### 4.10. Statistics

Data are presented as mean ± standard error of the mean (SEM) from 3 or more independent experiments. Data analysis was performed using one-way ANOVA followed by Tukey’s post hoc multiple comparison test (GraphPad software, Inc., San Diego, CA, USA). A probability value of *p* < 0.05 was considered statistically significant.

## Figures and Tables

**Figure 1 ijms-23-02388-f001:**
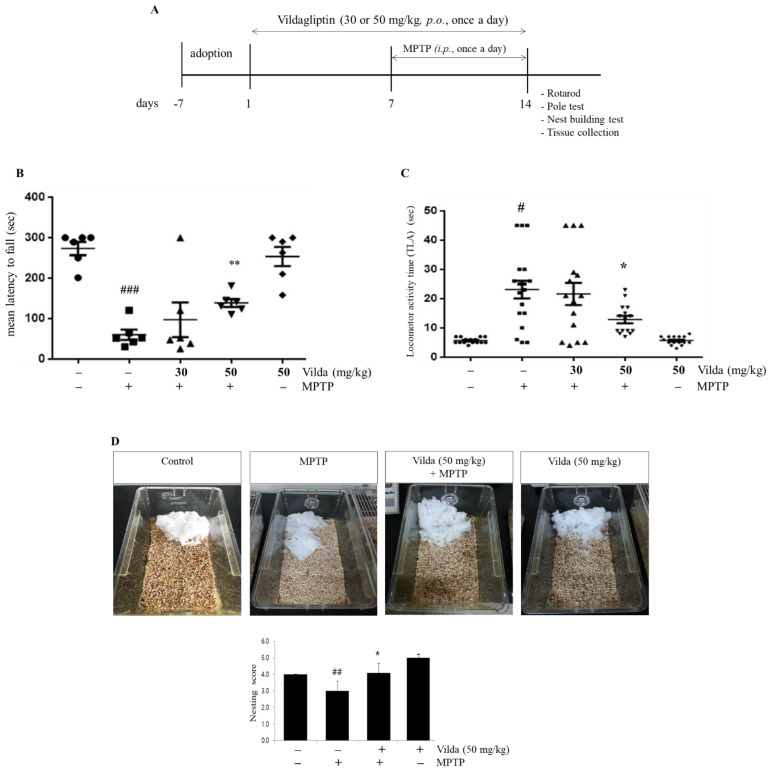
Vildagliptin inhibited the MPTP-induced motor impairments in mice. (**A**) Schematic representation of the treatment and behavioral study schedule for animal experiments. Mice were treated with vildagliptin (30 or 50 mg/kg, *p.o.*) or vehicle once per day for seven days. After that, mice were co-treated with MPTP (30 mg/kg, *i.p.*) or vehicle for another seven consecutive days. Motor function was assessed using (**B**) the rotarod test, (**C**) pole test, and (**D**) nest building test. All behaviors were analyzed on the 14th day after the last injection of MPTP. All values represent the mean ± SEM for 6 mice in each group. ^#^
*p* < 0.05, ^##^
*p* < 0.01, ^###^
*p* < 0.001 compared to the control group, * *p* < 0.05, ** *p* < 0.01 compared to the MPTP group.

**Figure 2 ijms-23-02388-f002:**
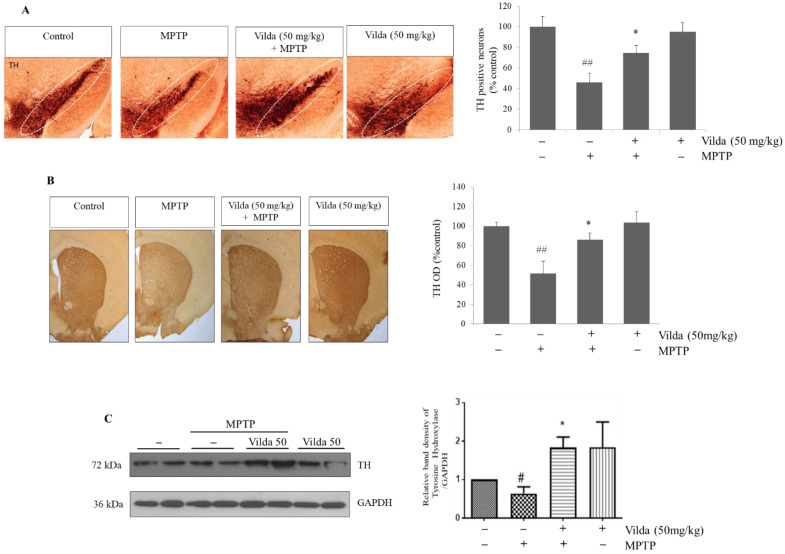
Vildagliptin protected against MPTP-induced loss of tyrosine hydroxylase (TH)-positive neurons in the SNpc and striatum. After the last injection of MPTP, the mice were sacrificed, brains were removed, and immunostaining was performed with anti-TH antibody. Representative pictures of dopaminergic neurons in the (**A**) SNpc (scale: 100×) and (**B**) striatum (scale: 40×) and their quantification. (**C**) The SNpc samples were isolated and used for immunoblots and quantification of relative TH protein level as shown. GAPDH expression level was as an internal control. Values are presented as mean fold-change ± SEM relative to the control (*n* = 6). ^#^
*p* < 0.05, ^##^
*p* < 0.01 compared to the control group, * *p* < 0.05 compared to the MPTP group.

**Figure 3 ijms-23-02388-f003:**
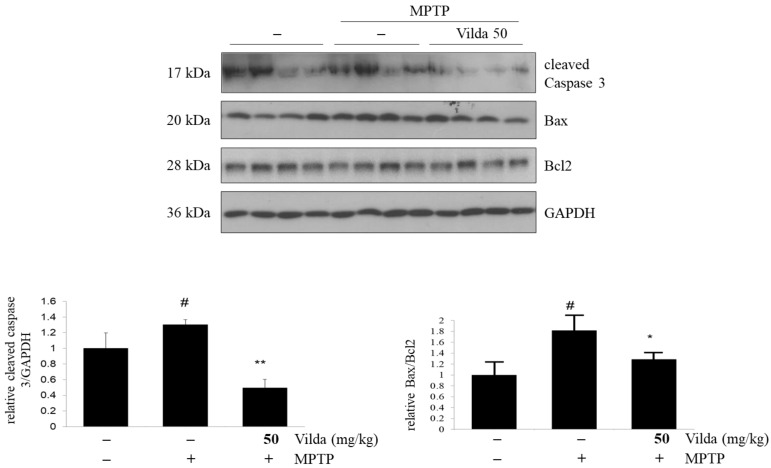
Vildagliptin attenuated MPTP-induced Bax/Bcl2 ratio and cleaved caspase-3 protein expression in mouse brains. Mice were treated with vildagliptin (50 mg/kg, *p.o.*) or vehicle once per day for seven days. After that, mice were co-treated with MPTP (30 mg/kg, *i.p.*) or vehicle for another seven consecutive days. After the behavioral tests, mice were sacrificed, and striatum tissues were collected. Striatum samples were used for immunoblots of caspase-3, Bax, and Bcl2 protein expression and densiometric quantification analysis. GAPDH expression level was used as an internal control. Values are presented as mean fold-change ± SEM relative to the control (*n* = 6). ^#^
*p* < 0.05 compared to the control group, * *p* < 0.05, ** *p* < 0.01 compared to the MPTP group.

**Figure 4 ijms-23-02388-f004:**
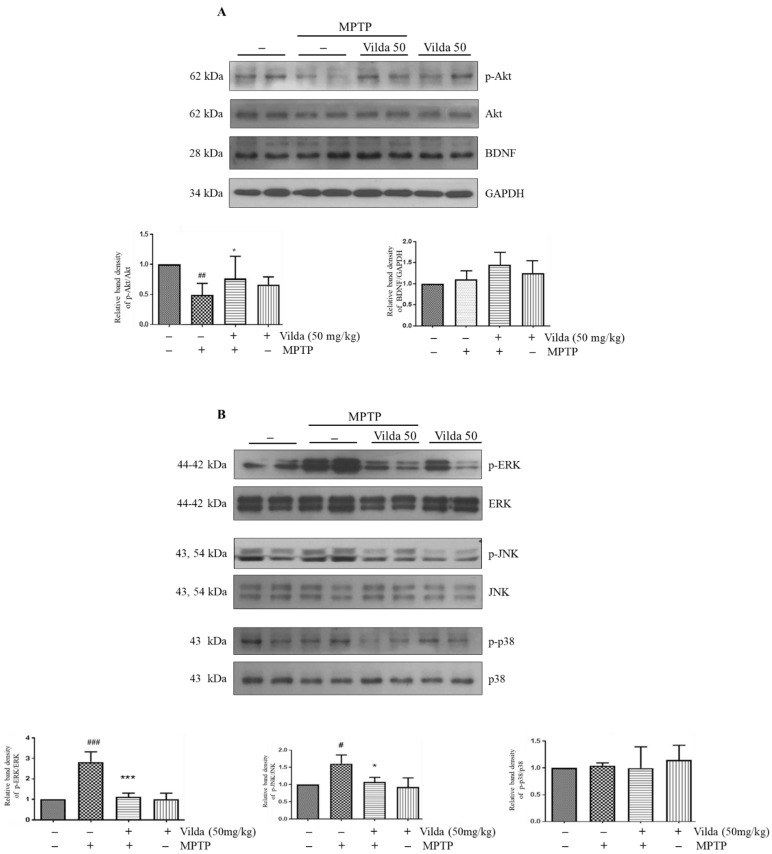
Vildagliptin restored MPTP-induced dephosphorylation of Akt and phosphorylation of ERK and JNK in the striatum. Striatum samples were used for immunoblots of Akt and MAPKs pathways. (**A**) Representative immunoblots of phospho-Akt, Akt, BDNF and GAPDH protein in mice striatum samples and their densiometric quantification. (**B**) Representative immunoblots of phospho-ERK, ERK, phospho-JNK, JNK, phospho-p38 and p38 protein in mice striatum samples and their densiometric quantification. Total Akt, ERK, JNK and p38 protein level was used as an internal control for their respective phosphorylated forms. Values are presented as mean fold-change ± SEM relative to the control (*n* = 6). ^#^
*p* < 0.05, ^##^
*p* < 0.01, ^###^
*p* < 0.001 compared to the control group, * *p* < 0.05, *** *p* < 0.001 with compared to the MPTP group.

**Figure 5 ijms-23-02388-f005:**
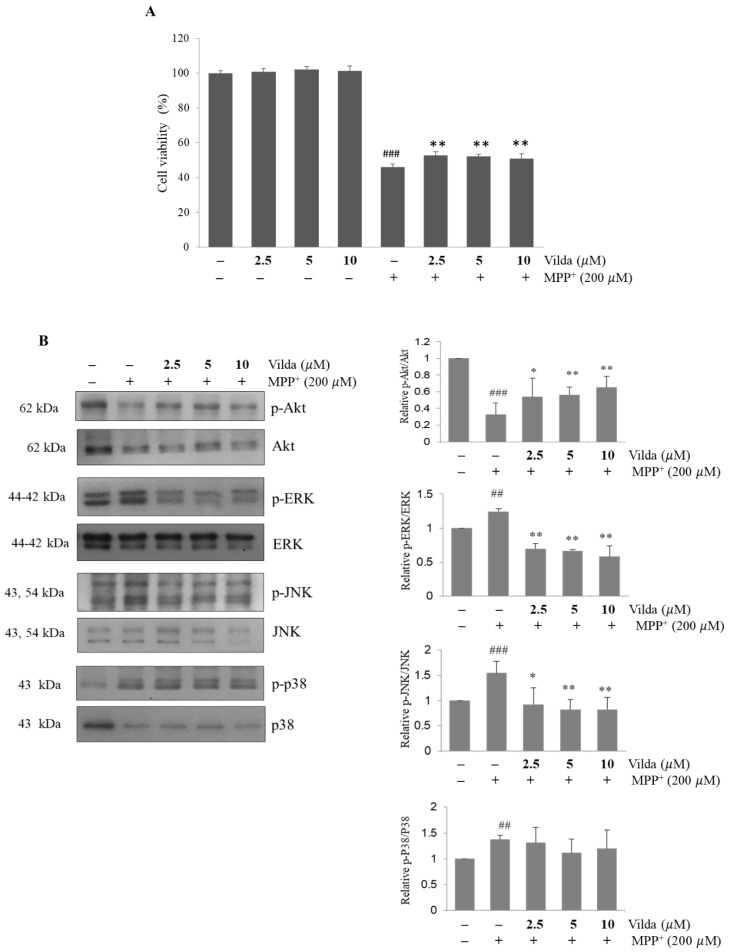
Vildagliptin maintained viability and phosphorylation of Akt, ERK and JNK in SH-SY5Y cells treated with MPP^+^. Vildagliptin (2.5, 5, and 10 μM) was pretreated for 2 h and exposed to MPP^+^ (200 μM) for 48 h. (**A**) Cell viability was measured by MTT assay, and (**B**) protein expressions of phospho-Akt, Akt, phospho-ERK, ERK, phospho-JNK, JNK, phospho-p38 and p38 were evaluated by immunoblot and densitometric analyses. MTT assay values are expressed as a percentage of the corresponding control cells and are presented as mean ± SD (*n* = 3). Total protein level of Akt, ERK, JNK or p38 was used as an internal control for their respective phosphorylated forms. Immunoblot densiometric values are presented as mean fold-change ± SEM relative to the control. ^##^
*p* < 0.01, ^###^
*p* < 0.001 compared to the control, * *p* < 0.05, ** *p* < 0.01 compared to the MPP^+^ group.

**Figure 6 ijms-23-02388-f006:**
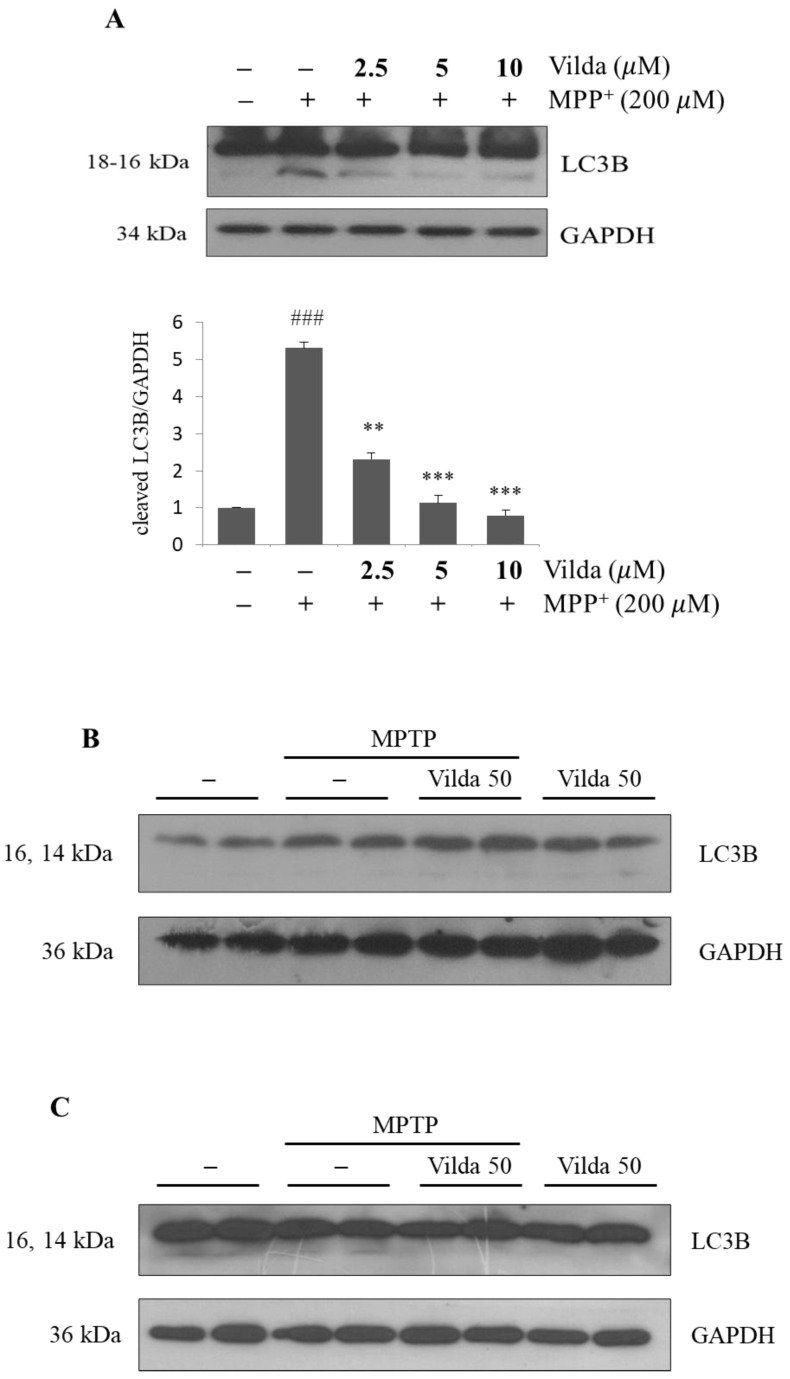
Vildagliptin attenuated MPP^+^-induced autophagy in SH-SY5Y cells. Vildagliptin (2.5, 5, and 10 μM) was pretreated for 2 h and exposed to a moderate concentration of MPP^+^ (200 μM) for 48 h. Cells were lysed, and protein expression was evaluated by immunoblot. (**A**) Immunoblot of LC3B protein expression and densiometric quantification. MPTP treated brain samples were collected after the behavioral tests. (**B**) SNpc and (**C**) striatum samples were used for immunoblots of LC3B protein expression. GAPDH expression level was as an internal control. Values are presented as mean fold-change ± SEM relative to the control (*n* = 3). ^###^
*p* < 0.001 with compared to control, ** *p* < 0.01, *** *p* < 0.001 with compared to MPP^+^ group.

## Data Availability

Data supporting the reported results are available on request from the corresponding author.

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
