# Peer review of "Neuroprotective Effects of the DPP4 Inhibitor Vildagliptin in In Vivo and In Vitro Models of Parkinson’s Disease"

_ijms, 2022, doi:10.3390/ijms23042388_

Round 1

Reviewer 1 Report

In this manuscript by Pariyar et al, authors have investigated the effects of vildagliptin on Perkinson’s disease and showed that vildagliptin provides protection against MPTP induced dopaminergic neurodegeneration.

This study has biological significance as the authors demonstrate that vildagliptin might have therapeutic effects in PD. The manuscript adds significant knowledge in the field. However, the text contains grammatical errors and text fonts are different. A, B, C in all the figures should be compiled in one figure for better clarity. 

Author Response

We highly appreciate your comments and suggestion. We checked grammar through English editing company before the submission, but it was not enough. In the revised manuscript, English grammar and spelling were checked by a specialist thoroughly. Typing error was carefully checked once again and corrected. As per your suggestion, in final paper version, A, B, C in all the figure will be compiled in one figure.   

Reviewer 2 Report

The present paper describes “Neuroprotective effects of the DPP4 inhibitor vildagliptin in 2 in-vivo and in-vitro models of Parkinson’s disease”. The work is interesting and good efforts have been lavished by authors. According to our consent, the quality of manuscript is publishable in Int. J. Mol. Sci. There are also some flaws that are needed to be improved before its publication in Int. J. Mol. Sci. Below are my comments on this manuscript.

  • The abstract part should be improved by adding more experimental data.
  • Introduction part must be improved to explain the current challenges being faced?
  • The authors are suggested to read these very important articles about dopamine detection as Parkinson diseases diagnosis, and cite them to strengthen the introduction part with more comparison among materials and techniques. Microchimica Acta (2019) 186:61 and Analytica Chimica Acta 1047 (2019) 197-207; Hazardous Mater. 2021, p. 126907
  • In conclusion, just present your results, novelty points
  • There should be comparative analysis of this study with other articles to highlight the advancement of this manuscript. A tabular form for comparison would be good.
  • There are many typographical errors in the paper. The manuscript must be rechecked for the better understanding.

Author Response

Reviewer #2:

The present paper describes “Neuroprotective effects of the DPP4 inhibitor vildagliptin in 2 in-vivo and in-vitro models of Parkinson’s disease”. The work is interesting and good efforts have been lavished by authors. According to our consent, the quality of manuscript is publishable in Int. J. Mol. Sci. There are also some flaws that are needed to be improved before its publication in Int. J. Mol. Sci. Below are my comments on this manuscript.

Comment # 1) The abstract part should be improved by adding more experimental data.

Answer #1) We highly appreciate your comments. As per suggestion, we have added more experimental data and reduced some introduction part at Abstract section in our reviewed manuscript . The changes we made are marked with yellow highlights in the revised manuscript.

“Abstract:

Parkinson’s disease (PD) is the second most common neurodegenerative disease, characterized by loss of dopaminergic neurons in the substantia nigra pars compacta (SNpc) of the midbrain. Restoration of nigrostriatal dopamine neurons has been proposed as a potential therapeutic strategy for PD. Because currently used PD therapeutics only help relieve motor symptoms and do not treat the cause of the disease, highly effective drugs are needed. Vildagliptin, a dipeptidyl peptidase 4 (DPP4) inhibitor, is an anti-diabetic drug with various pharmacological properties including neuroprotective effects. However, the detailed effects of vildagliptin against PD are not fully understood. We investigated the effects of vildagliptin on PD and its underlying molecular mechanisms using a 1-methyl-4-phenyl-1,2,3,6-tetrahydropyridine (MPTP)-induced mouse model and a 1-methyl-4-phenylpyridium (MPP+)-induced cytotoxicity model. Vildagliptin (50 mg/kg) administration significantly attenuated MPTP-induced motor deficits as evidenced by rotarod, pole, and nest building tests. Immunohistochemistry and western blot analysis revealed that vildagliptin increased tyrosine hydroxylase-positive cells in the SNpc and striatum, which was reduced by MPTP treatment. Furthermore, vildagliptin activated MPTP-decreased PI3k/Akt and mitigated MPTP-increased ERK and JNK signaling pathways in the striatum. Consistent with signaling transduction in the mouse striatum, vildagliptin reversed MPP+-induced dephosphorylation of PI3K/Akt and phosphorylation of ERK and JNK in SH-SY5Y cells. Moreover, vildagliptin attenuated MPP+-induced conversion of LC3B-II in SH-SY5Y cells, suggesting its role in autophagy inhibition. Taken together, these findings indicate that vildagliptin has protective effects against MPTP-induced motor dysfunction by inhibiting dopaminergic neuronal apoptosis, which is associated with regulation of PI3k/Akt, ERK, and JNK signaling transduction. Our findings suggest vildagliptin as a promising repurposing drug to treat PD.

Comment #2) Introduction part must be improved to explain the current challenges being faced?

The authors are suggested to read these very important articles about dopamine detection as Parkinson diseases diagnosis, and cite them to strengthen the introduction part with more comparison among materials and techniques. Microchimica Acta (2019) 186:61 and Analytica Chimica Acta 1047 (2019) 197-207; Hazardous Mater. 2021, p. 126907

Answer #2) As reviewer’s kind suggestions, we have added the explanation of current challenges in PD diagnosis in the introduction section with the suggested references. The changes we made are marked with yellow highlights in the revised manuscript

“Introduction : Parkinson’s Disease (PD) is the second most common neurodegenerative disorder after Alzheimer’s disease [1]. Clinically, PD symptoms are divided into two categories, motor and non-motor symptoms. The motor symptoms are resting tremor, bradykinesia, rigidity, and posture instability [2, 3]. The main pathological hallmark of PD is progressive death of dopaminergic neurons in the substantia nigra pars compacta (SNpc) and striatum in the brain. The pathology responsible for the clinical conditions is accompanied by degeneration of dopaminergic neurons in the SNpc [4] and thereby reduction of dopamine level in the striatum [5, 6]. Cardinal symptoms of PD are observed only after the dopamine level in the striatum is decreased by 60-80% [7]. Therefore, restoring dopamine/acetylcholine balance is the primary goal of pharmacotherapy treatment to mitigate symptoms associated with PD [8]. Recently, real-time monitoring of dopamine release based on nanocomposites has been studied as a potential diagnostic tool for PD [9, 10].

Comment #3) In conclusion, just present your results, novelty points.

Answer #3)  As reviewer’s kind suggestions, we have made some changed at conclusion section. The changes we made are marked with yellow highlights in the revised manuscript

In conclusion, our findings suggest that vildagliptin restored MPTP-induced dopaminergic neurodegeneration in the SNpc and striatum of the mouse brain and ameliorated MPTP-induced motor dysfunction. Vildagliptin also reduced MPP+-induced neurocytotoxicity in SH-SY5Y cells. Furthermore, vildagliptin inhibited MPTP/MPP+-induced dephosphorylation of Akt and phosphorylation of JNK and ERK in in-vivo and in-vitro PD models. Moreover, vildagliptin mitigated MPP+-induced autophagy in SH-SY5Y cells. Our data demonstrate that vildagliptin might have the therapeutic effects in PD by inhibiting dopaminergic neuronal apoptosis in the nigrostriatal system, suggesting the potential of vildagliptin as a repurposed drug against PD.

Comment #4) There should be comparative analysis of this study with other articles to highlight the advancement of this manuscript. A tabular form for comparison would be good.

Answer #4)  We highly appreciate your comments. I aggree with you tabular form for comparison is a great idea. In discussion section, we have discussed and compared our vildagliptin data with published data of other researchers. We will definitely include tabular comparison in our other projects. Thank you for your suggestion.

Comment #5) There are many typographical errors in the paper. The manuscript must be rechecked for the better understanding.

Answer #5) Thank you for your suggestion. In the revised manuscript, English grammar and spelling were checked by a specialist. Typographical errors were carefully checked once again and corrected.

Reviewer 3 Report

The paper adds further positive arguments for repositioning of dipeptidyl peptidase-4 inhibitors, among them Vildagliptin, as potential neuroprotective agents. The experimental design and methods are appropriate. The conclusions are supported by the results.  

Author Response

Thank you so much to have reviewed our manuscript. We highly appreciate your comments.

Round 2

Reviewer 1 Report

The authors have addressed my concerns. The manuscript can be accepted for publication.